# Evaluation of "Caserotek" a low cost and effective artificial blood-feeding device for mosquitoes

**Helvio Astete[1], Verónica Briesemeister[2], Cesar Campos[2,3], Angel Puertas[1], Thomas W. Scott[2], Víctor López-Sifuentes[1], Ryan Larson[1], Michael Fisher[1], Gissella M. Vásquez[1], Karin Escobedo-Vargas[1], Amy C. Morrison ⓘ [4] ***

**1** U.S. Naval Medical Research Unit No. 6, Peru, **2** Department of Entomology, University of California, Davis, United States of America, **3** Vysnova Partners Incorporated, Landover, Maryland, United States of America, **4** Department of Pathology, Microbiology, and Immunology, School of Veterinary Medicine, University of California, Davis, United States of America

\* amy.aegypti@gmail.com

**Data Availability Statement:** The authors confirm that all data underlying the findings are fully available without restriction. All relevant data are

## Abstract

Entomological research studies on mosquito vector biology, vector competence, insecticide resistance, dispersal, and survival (using mark-release-recapture techniques) often rely on laboratory-reared mosquito colonies to produce large numbers of consistently reared, aged, and sized mosquitoes. We developed a low-cost blood feeding apparatus that supports temperatures consistent with warm blooded animals, using commonly available materials found in low resource environments. We compare our system ("Caserotek") to Hemotek and glass/membrane feeding methods. Two experiments were conducted with *Aedes aegypti* (Linnaeus 1762) and one with *Anopheles darlingi* (Root 1926) (Diptera: Culicidae); 3 replicates were conducted for each experiment. *Aedes aegypti* female mosquitoes were provided chicken blood once per week for 30 min (Experiment #1) for 14 days or 1 hour (Experiment #2) for 21 days. *Anopheles darlingi* were fed once for 1 hour (Experiment #3). Blood-feeding rates, survival rates, and egg production were calculated across replicates. Caserotek had a significantly higher 30-min engorgement rate (91.1%) than Hemotek (47.7%), and the glass feeder (29.3%) whereas for 1-hour feeding, Hemotek had a significantly lower engorgement rate than either of the other two devices (78% versus 91%). Thirty-day survival was similar among the feeding devices, ranging from 86% to 99%. Mean egg production was highest for the Caserotek feeder (32 eggs per female) compared to the glass feeder and Hemotek device (21–22 eggs per female). Our new artificial feeding system had significantly higher blood feeding rates than for more expensive artificial systems and was equivalent to other fitness parameters. Caserotek only requires the ability to boil water to maintain blood temperatures using a Styrofoam liner. It can be easily scaled up to large production facilities and used under austere conditions.

within the paper and its Supporting information files. Primary data sets used for data analysis are provided in the supplementary information files.

**Funding:** This study was funded by the U.S. National Institute of Allergy and Infectious Diseases (NIH/NIAID) award number P01 AI098670 (TWS). ACM receives salary support from award U01AI151814 from the National Institute of Allergy and Infectious Diseases of the National Institutes of Health. Further support was provided by Military Infectious Diseases Research Program (MIDRP) Proposal U0501_17_AF_CS_OC. The funders had no role in study design, data collection and analysis, decision to publish, or preparation of the manuscript.

**Competing interests:** The authors have declared that no competing interests exist.

## Author summary

To carry out studies on mosquitoes including if they can transmit viruses, are resistant to insecticides, how far they can fly, and how long they live, scientists raise mosquitoes in laboratories where they must feed them blood to produce large numbers of similar sized mosquitoes. We developed a low-cost device made with materials available at most hardware stores throughout the world. We compare "Caserotek" to other commercially available blood feeding methods, thorough two experiments with *Aedes aegypti* and one with *Anopheles darlingi*. We fed *Aedes aegypti* female mosquitoes on chicken blood once per week for 30 min (Experiment #1) for 14 days or 1 hour (Experiment #2) for 21 days. We fed *Anopheles darlingi* for 1 hour (Experiment #3). We measured how well mosquitoes fed on blood (feeding rates), how well the blood fed mosquitoes survived, and how many eggs the mosquitoes laid. *Aedes aegypti* mosquitoes fed for 30 minutes, fed best on Caserotek (91.1%) compared to 47.7% and 29.3% on Hemotek and the glass feeder, respectively. When *Aedes aegypti* fed for 1-hour feeding rates, were good (91%) on Caserotek and the glass feeder, but lower on Hemotek (78%). Thirty-day survival was similar among the feeding devices, ranging from 86% to 99%. Average egg production was highest for the Caserotek feeder (32 eggs per female) compared to the glass feeder and Hemotek device (21–22 eggs per female). Caserotek performed well compared to other more expensive feeding devices. Caserotek only requires the ability to boil water to maintain blood temperatures using a Styrofoam liner. It can be easily scaled up to large production facilities and used under austere conditions.

## Introduction

Entomological research studies on vector biology, vector competence, development of new repellents, insecticide resistance, dispersal, and survival (using mark-release-recapture techniques) often rely on laboratory-reared mosquito colonies for the production of large numbers of healthy, consistently sized and aged mosquitoes [1]. Female mosquitoes require a blood meal for egg production necessitating a blood source to maintain colonies in the laboratory. Although mosquito colonies have used live animals for blood feeding; artificial feeding systems remove the administrative and logistical burdens associated with animal use protocols, which in turn reduces research costs [2]. Effective artificial blood feeding systems must contain blood in a vessel or substrate while simultaneously providing the mosquitoes access to a blood source via membrane (e.g., animal tissues, Parafilm-M films, or collagen membranes). Additionally, artificial feeding systems must regulate the temperature of the blood as heat is one of the primary cues that mosquitoes use to locate hosts [1,3]. In recent reviews, over 20 devices have been described for mosquitoes [1,3,4]. Two of the most commonly used systems include the glass membrane feeder first developed by Rutledge et al. [5,6] and the commercially available Hemotek system [7,8]. The glass membrane feeder is characterized by an inner chamber of heat-resistant glass, surrounded by a cylindrical water jacket with inlet and outlet tubes connected to rubber tubing that holds circulated water held at a constant temperature (e.g. water bath) and can be used with a variety of membranes. Multiple glass feeders can be connected to a single water bath and multiple systems can be used simultaneously to increase the number of mosquito cages fed at one time. Hemotek can feed up to 5–6 mosquito cages at once by heating blood with an electrical device but can cost up to $3,000 and is also time consuming and cumbersome to use [1,3,9]. We have developed a low-cost blood feeding apparatus that does not require an external power source (as do Hemotek, or glass feeders that require connection to a

water bath) to maintain blood at a temperature consistent with warm blood animals. Our device uses commonly available materials found in low resource environments and can be easily scaled to large production facilities. We describe and evaluate our handmade feeding system called "Caserotek" by comparing to Hemotek and traditional glass/membrane feeding systems that are more expensive and difficult to procure. This novel product could be useful for the maintenance of colonies of a large range of important mosquito vectors.

## Methods and materials

### Insectaries

Our experiments were conducted in Iquitos City located within the Amazon rainforest (73.2'W longitude, 3.7˚S latitude, 120 m above sea level) in the Department of Loreto, Northeastern Peru. Experiments were conducted in two insectary facilities. The first was a field insectary constructed in a local household and maintained by the University of California, Davis (Fig 1) [10], dedicated to ongoing vector competence studies for *Aedes aegypti* [10]. An area was screened off with a double entrance to prevent escape of mosquitoes and contained a single air conditioner that was used to regulate the room's temperature. The second was the Naval Medical Research Unit No. 6 (NAMRU-6) Iquitos Insectary facility with an *Anopheles darlingi* colony but with some space dedicated to *Ae. aegypti* [11].

### Measurement of environmental conditions

Daily maximum and minimum temperature and relative humidity were measured with a Hygrometer/Thermometer (Thomas Traceable Hygrometer/Thermometer, Thomas Scientific). Mosquitoes were maintained at 27˚C. The insectary had windows that let in natural light and room lights were turned on from approximately 0800 to 1700 each day with the photoperiod in Iquitos averaging 12:12 light/dark (range between minimum and maximum daylight is approximately 1 hour). Iquitos city is surrounded by tropical rainforest, with an average daily temperature of 25˚C and an average annual precipitation of 2.7 meters. We measured the temperature and relative humidity during each feed.

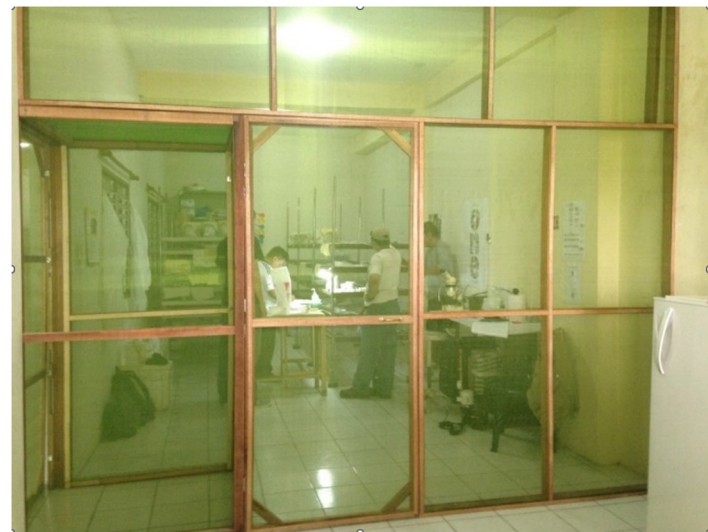

**Fig 1. University of California Davis Field Insectary.**

## Mosquitoes

*Aedes aegypti* used in experiments were reared in the UC Davis insectary described previously, whereas *An. darlingi* were reared at the NAMRU-6 insectary. *Aedes aegypti* were derived from larvae, pupae from routine *Ae. aegypti* surveys conducted approximately 1-month before the experiments, or from eggs collected using ovitraps placed in ten geographically distinct neighborhoods within Iquitos. Egg papers were allowed to dry for 1–2 days then stored in ziplock bags in plastic containers within 2 weeks. Eggs were placed in 1L of a water/tea (Collins Cinnamon tea) infusion at 40°C for 24 hours. Larvae were placed in white plastic pans (26.5 by 16.5 cm) containing 1000 mL of tap water. Larvae were provided a combination of wheat powder mixed with commercial fish food daily until pupation. Pupae were transferred to 1 pint plastic containers and placed in metal cages and allowed to emerge. Thus, the mosquitoes represent a cross-section of $F_0$ adults from immature stages collected from the field.

For experiments using *Ae. aegypti* a total of 100 female and 50 male pupae were placed into 1-gallon plastic cages from larger metal cages where pupae emerged. These cages were covered with a moist towel to maintain humidity but were not provided with sugar or water. Mosquitoes 3–5 days of age were offered blood using three different feeding apparatuses. After feeding, sugar cubes and a water pledget were placed on the top of the cages until approximately 24 hours before subsequent blood feeding on day 7, 14, and 21.

*Anopheles darlingi* used in experiments were from generation $F_{62}$ from a colony maintained by NAMRU-6 where the mosquitoes are provided with 10% sucrose solution ad libitum [11]. Before feeding, 100 female mosquitoes that were between 3–7 days post-emergence, were aspirated and placed into 1-gallon plastic containers without sugar for approximately 24 hours. The cages were again provided with 10% sucrose solution ad libitum until 24 hours before providing a single blood meal for three separate cohorts of mosquitoes.

## Blood

We utilized chicken blood collected in EDTA tubes from a local butcher, where blood was available for purchase for all experiments.

## Description of Caserotek blood feeding apparatus (Figs 2 and S1–S4)

The device uses a plastic urine collection container (85 mm x 64 mm). A plastic tube (75 mm) is run through the base of the cup flush with the lid, so that blood will flow to the space between the top of the lid and the 10 mm lid lip. Two holes are cut in the base of the cup to add hot water (18 mm) and to hold a tube to pour blood (10 mm). The interior of the cup is lined with the Styrofoam from an 8 oz coffee cup. A Teflon film (plumbers' tape) covers the elevated lip of the lid allowing a narrow space between the lid top and the Teflon film. Mosquitoes feed easily through the film. Hot water is placed in the cup, then a minimum of 2 ml of blood is added to the tube. Hot water is replaced at 20-minute intervals. Prior to feeding, our laboratory technician rubbed the membrane area on exposed skin with sweat to stimulate feeding. A standard operating procedure is provided in supplementary information (S1 File).

## Study design

We conducted two sets of experiments for *Aedes aegypti* and one for *Anopheles darlingi*. During each experiment, three blood feeding devices (Hemotek, glass membrane [3.8 cm outer diameter] feeder, and Caserotek) were tested simultaneously. The Hemotek and glass membrane feeders used parafilm membranes whereas, Caserotek utilized Teflon tape as a membrane. All membrane types were rubbed with sweat from the same laboratory technician to

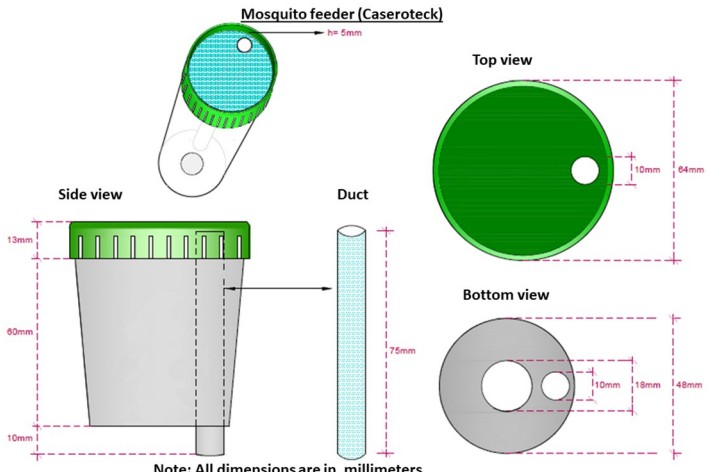

**Fig 2. Description of Caserotek Device.**

stimulate feeding. Each trial included three replicates. A total of 100 females and 50 males were transferred from larger metal cages where pupae emerged to 1 gallon plastic cages. These cages were covered with a moist towel to maintain humidity but not provided sugar or water. Three-to-five-day-old mosquitoes were offered chicken blood using three different feeding apparatuses. After feeding, sugar cubes and a water pledget were placed on the top of the cages until approximately 24 hours before subsequent blood feeding.

For the *Ae. aegypti* experiments (#1 and #2), the blood-feeding duration, number of feeds, length of observation, and number of egg collections differed between experiments; however, blood feeding rates (engorged mosquitoes), survival rates, and egg production were compared across blood feeding devices in both experiments. In contrast, for *An. darlingi* experiments, blood feeding rates, egg hatch rates, and adult emergence were evaluated once on 7 days after a blood meal. Below we describe the details of individual experiments.

*Experiment 1 (11-September-2017–2 October-2017, see* S5 Fig*)*: *Aedes aegypti* were provided blood for 30 minutes on day 1, 7, and 14 and subsequently, eggs were collected on day 7, 14, and 21 to evaluate egg production. After blood feeding, all unfed female mosquitoes were eliminated. Adult mortality was evaluated each day, when oviposition sites are provided to mosquitoes.

*Experiment 2 (16-January-2020–24-February-2020, see* S6 Fig*)*: *Aedes aegypti* were provided blood for 1 hour on day 1, 7, 14 and 21 and subsequently, eggs collected on day 7, 14, 21, and 30 to evaluate egg production. Mosquitoes were transported to and from the UC Davis field insectary to the NAMRU-6 insectary on each feeding day. Transport of mosquito cages between insectaries was done approximately 1 hour prior to initiating feeding. Mosquito cages were placed in Styrofoam coolers containing moist paper towels, transferred to an air-conditioned vehicle for the 5-minute drive to the NAMRU-6 insectary where the cages were removed from the boxes in the same room as the feeding apparatus and the mosquitoes allowed to acclimatize for a minimum of 30 minutes before providing blood. After blood feeding all unfed females remained in the cage. Adult mortality was evaluated each day.

*Experiment 3 (11-January-2018–22 February-2018, see* S7 Fig*)*: *Anopheles darlingi* were allowed to feed for 1 hour on day 1; eggs were collected approximately 1 week later. Although

overall egg production could not be counted accurately, a subset of eggs was evaluated for hatch rate and adult emergence rate.

### Statistical analysis

To identify differences in engorgement rates, egg production per female mosquito, and survival rates we used analysis of variance (ANOVA) with the general linear models procedure (PROC GLM) of SAS [12]. Models were constructed for each of dependent variables separately for each experiment described above. Independent variables included in the models were day/week (day 1, 7, 14, and 21) of the experiment, replicate (A, B, C), and device (glass feeder, Hemotek, or Caserotek) used. We also considered the day*device interaction. Least square means (LSMeans) were used to test differences among mean rates within main effects and interactions terms; the significance level was adjusted based on pre-planned comparisons. In experiment #1, data points with less than 10 mosquitoes were removed the analysis. Complete model results for analyses with and without all data points are included in supplementary information (S2 File).

## Results

### Experiment #1

**Feeding rates.** During the 30-minute blood-feeding period female *Ae. aegypti*, feeding rates were significantly different between the experimental day and feeding apparatus (p <0.0001, Table 1). Additionally, there was a significant day*device interaction (p = 0.0031); these variables accounted for 91% (r-square) of the data variation in our GLM model (Table 1, S2 File for full model results). The Caserotek device consistently had the highest engorgement rates, >88% across all days or least square mean of 91.4% (stderr = 3.6) compared to 62.0% and 50.0% for the Hemotek and glass feeder, respectively (LS Means; p ≥ 0.0001) across day 1, 7 and 14. Feeding rates were significantly lower on day 1 than other days for both the glass feeder and Hemotek devices (Fig 3).

Egg production and survival. Because unfed females were removed in experiment #1 after being offered a blood meal, the number of mosquitoes evaluated per device/week/replicate for both egg production and survival was not well balanced because of the lower feeding rates observed in the Hemotek and glass feeders. For example, one glass feeder replicate only had two individual mosquitoes remained for three time points, and one Hemotek replicate only had 14 mosquitoes. The average number of engorged mosquitoes included was 83 (range, 64 to 97) for Caserotek, 28 (range, 14 to 36) for Hemotek, and only 16 (range, 2 to

**Table 1. Full General Linear Model (GLM) results for effect of feeding day and device on engorgement rate after exposing mosquitoes to blood for 30 minutes.**

| Source | DF | SS | Mean Square | F Value | Pr>F |
|---|---|---|---|---|---|
| Model | 8 | 18193 | 2274 | 19.38 | <0.0001 |
| Error | 16 | 1878 | 117 | | |
| Corrected Total | 24 | 20071 | | | |
| R-Square | Coeff Var | Root MSE | Mean | | |
| 0.91 | 15.95 | 10.83 | 67.93 | | |
| Source | DF | Type III SS | Mean Square | F Value | Pr > F |
| Day | 2 | 6603 | 3301 | 28.13 | <0.0001 |
| Device | 2 | 7399 | 3699 | 31.52 | <0.0001 |
| Day*Device | 4 | 2937 | 734 | 6.26 | 0.0031 |

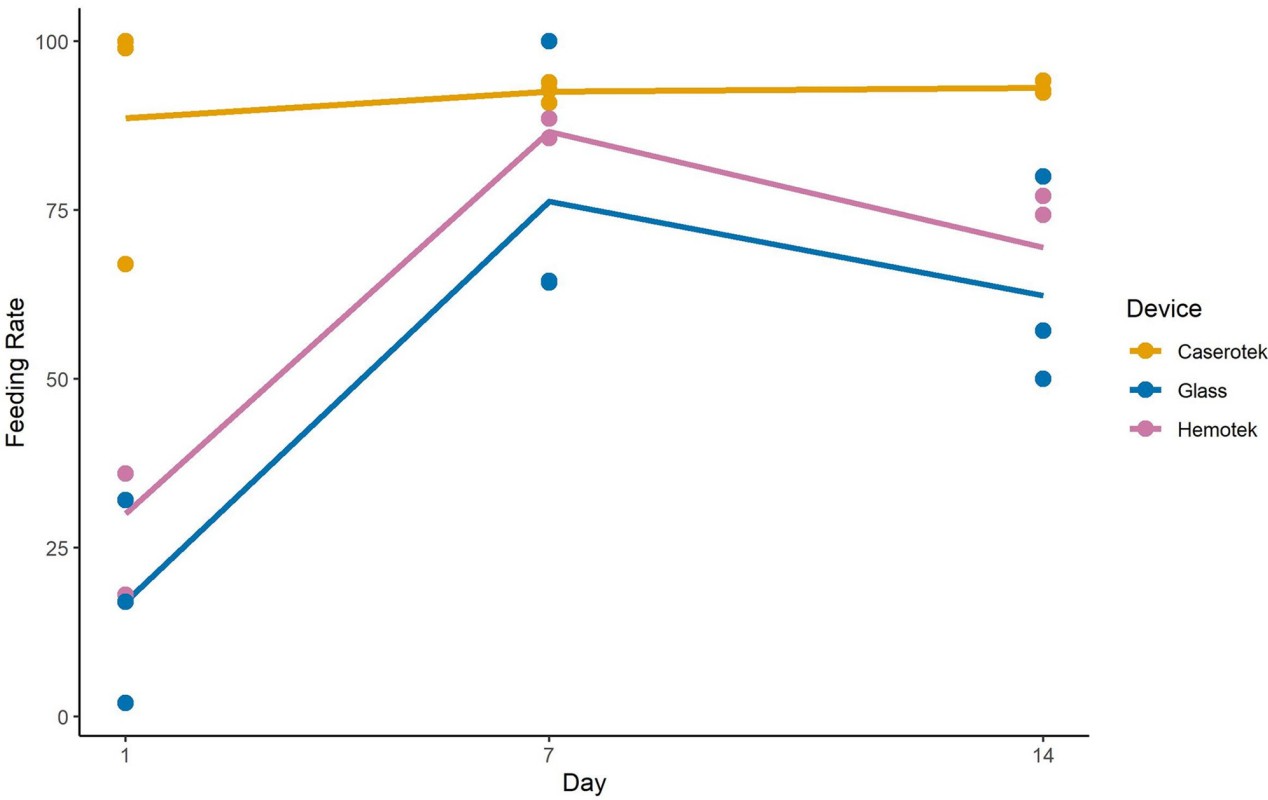

**Fig 3. Interaction Plot feeding rates by day and device for *Aedes aegypti* experiment 1.** Closed circles represent individual replicates whereas lines show mean feeding rates.

31) for the glass feeder. The average number of eggs laid per engorged female was highest for the glass feeder (58.9±42.4), followed by Hemotek (36.3±12.2), and finally Caserotek (27.9 ±8.1); these differences were not statistically significant (p = 0.11, S2 File). Survival rates of engorged mosquitoes were not statistically different, ranging from 84.4% (±8.8%) using Hemotek, 89.0% (±11.8%) for the glass feeder, and 90.1% (±5.5%) for Caserotek (p = 0.81, S2 File and S8 Fig).

### Experiment #2

**Feeding rates.**   During the 1-hour blood-feeding period for female *Ae. aegypti*, feeding rates were significantly different between the experimental day and feeding apparatus (p <0.0001, Table 2, S2 File). Additionally, there was a significant day*device interaction (p<0.0001, Table 2); these variables accounted for 95% (r-square) of the data variation in our GLM model. Engorgement rates were significantly lower on the first feed (LS mean = 58%) than subsequent feeds on day 7, 14, and 21 (LS mean = 95–98%) (p<0.0001). Engorgement rates were significantly lower for Hemotek (LS mean = 78%) than either the Caserotek device (LS mean = 92%) or glass feeder (LS mean = 91%) (p<0.0001). The lower feeding rate observed using the Hemotek device can be explained by differences on the day 1 feed; only 26% fed on Hemotek compared to 72% with Caserotek and 75% with the glass feeder (p<0.0001) (Fig 4).

**Table 2. Full General Linear Model (GLM) results for effect of feeding day and device on engorgement rate after exposing mosquitoes to blood for 1 hour.**

| Source | DF | SS | Mean Square | F Value | Pr>F |
|---|---|---|---|---|---|
| Model | 11 | 14751 | 1341 | 40.86 | <0.0001 |
| Error | 24 | 788 | 33 | | |
| Corrected Total | 35 | 15539 | | | |
| R-Square | Coeff Var | Root MSE | Mean | | |
| 0.95 | 6.58 | 573 | 87.06 | | |
| Source | DF | Type III SS | Mean Square | F Value | Pr > F |
| Day | 3 | 10175 | 3392 | 103 | <0.0001 |
| Device | 2 | 1383 | 691 | 21 | <0.0001 |
| Day*Device | 6 | 3193 | 532 | 16 | <0.0001 |

**Survival.** Survival rates were very high overall and nearly identical for the three devices, ranging from 99.7% (±11.8%), 99.8% (±0.48%), and 99.9% (±0.31%) for Hemotek, glass feeder, and Caserotek devices, respectively.

**Egg Production.** Egg Production varied significantly by experimental week (p = 0.0062) and the device used (p<0.0001) (Table 3 and S2 File). Egg production was highest in week 3 (see Fig 4). Overall, *Ae. aegypti* fed using the Caserotek device produced significantly more eggs per female (32±1.6) than either Hemotek (21±1.6) or the glass feeder (22±1.6) (LS means, p≤0.0002) (Fig 5).

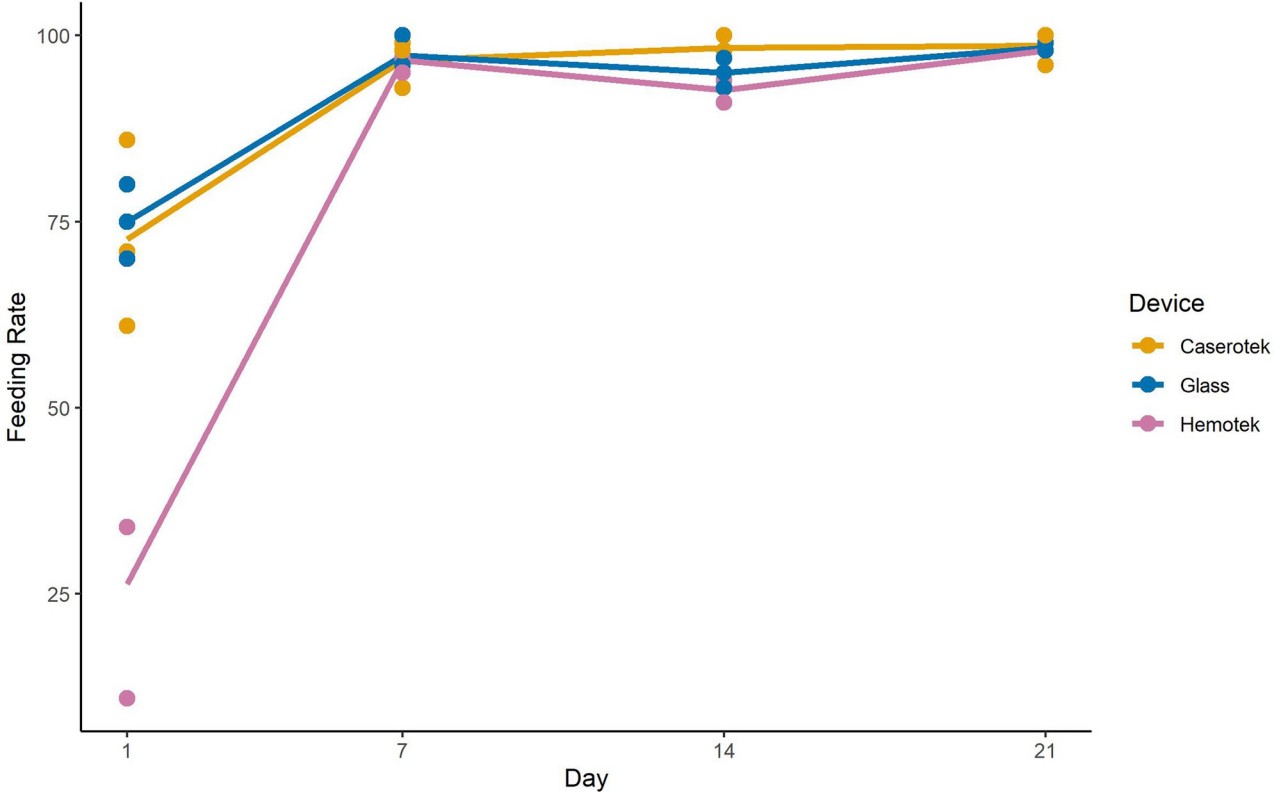

**Fig 4. Interaction Plot feeding rates by day and device for *Aedes aegypti* experiment 2.** Open circles represent individual replicates whereas lines show mean feeding rates.

**Table 3. Full General Linear Model (GLM) results for effect of week of egg collection post initial blood feed (1 hour duration) and device on the number of eggs laid per mosquito.**

| Source | DF | SS | Mean Square | F Value | Pr>F |
|---|---|---|---|---|---|
| Model | 5 | 1330 | 266 | 41 | <0.0001 |
| Error | 30 | 912 | 30 | | |
| Corrected Total | 35 | 2242 | | | |
| R-Square | Coeff Var | Root MSE | Mean | | |
| 0.59 | 22 | 6 | 25 | | |
| Source | DF | Type III SS | Mean Square | F Value | Pr > F |
| Week | 3 | 458 | 153 | 5 | 0.0062 |
| Device | 2 | 872 | 436 | 14 | <0.0001 |

## Experiment 3

**Feeding rate.** During the 1-hour period female *An. darlingi* were provided blood, feeding rates were relatively consistent over experimental week and device used. There was strong evidence of a week*device interaction (df = 4, f-value 4.91, p = 0.0089); these variables accounted for 62% (r-square) of the data variation in our GLM model (S2 File). Although not statistically significant, engorgement rate for Caserotek (81%± 12.0%) was higher than observed for either the glass feeder (72%± 10.8%) or Hemotek devices (75%± 11.0%). Only during week 3 was there a significant difference between feeding rates for Caserotek (LS mean = 89%) and Hemotek (LS mean = 61%) while the rate for the glass feeder was in between (LS mean = 74%).

**Egg Production and hatch rate.** Eggs laid per *An. darlingi*, female ranged from 11.6 to 14.3 eggs/female in the Hemotek and Caserotek devices, respectively. Hatch rates were lower for the glass feeder (69%) and highest for Hemotek (83%); however, none of these differences were statistically significant. Full model results are available in supplementary information (S2 File).

## Discussion

Herein, we compared our Caserotek blood feeding system to the more commonly used glass membrane feeder and Hemotek blood feeding systems for mosquito colony maintenance. Overall, the Caserotek device either performed as well as or outperformed the other devices in all parameters measured for both *Ae. aegypti* and *An. darlingi*. Feeding rates were significantly higher for *Ae. aegypti* in all experiments and feeding happened more quickly than for the other devices. Additionally, *Ae. aegypti* produced more eggs per female on average than for the other devices.

Overall, feeding, fecundity, and survival rates observed for Caserotek, were consistent with those observed for other studies and devices, which varied widely depending on the device, membranes, and methods used. For example, use of a Glytube feeder engorgement rates ranged from 38% for a sheep intestine membrane to 63% for plumbers tape, the same membrane used with the Caserotek device [13]. In another study comparing feeding rates of *An. coluzzii* on Hemotek and glass feeding devices across different membranes, average feeding rates were 49% and 42%, respectively [14]. Of the published devices available, the most similar to Caserotek was that developed by Siria et al. [15] which used both Styrofoam cups and polytetrafluoroethylene (PTFE) tape as a membrane. Their device feeding rates were 100% for *Ae. aegypti*, 99% for *An. gambiae*, and 86% for *An. arabiensis* for a 20-minute feeding period. Estimates of fecundity were approximately twice as high for Caserotek as the

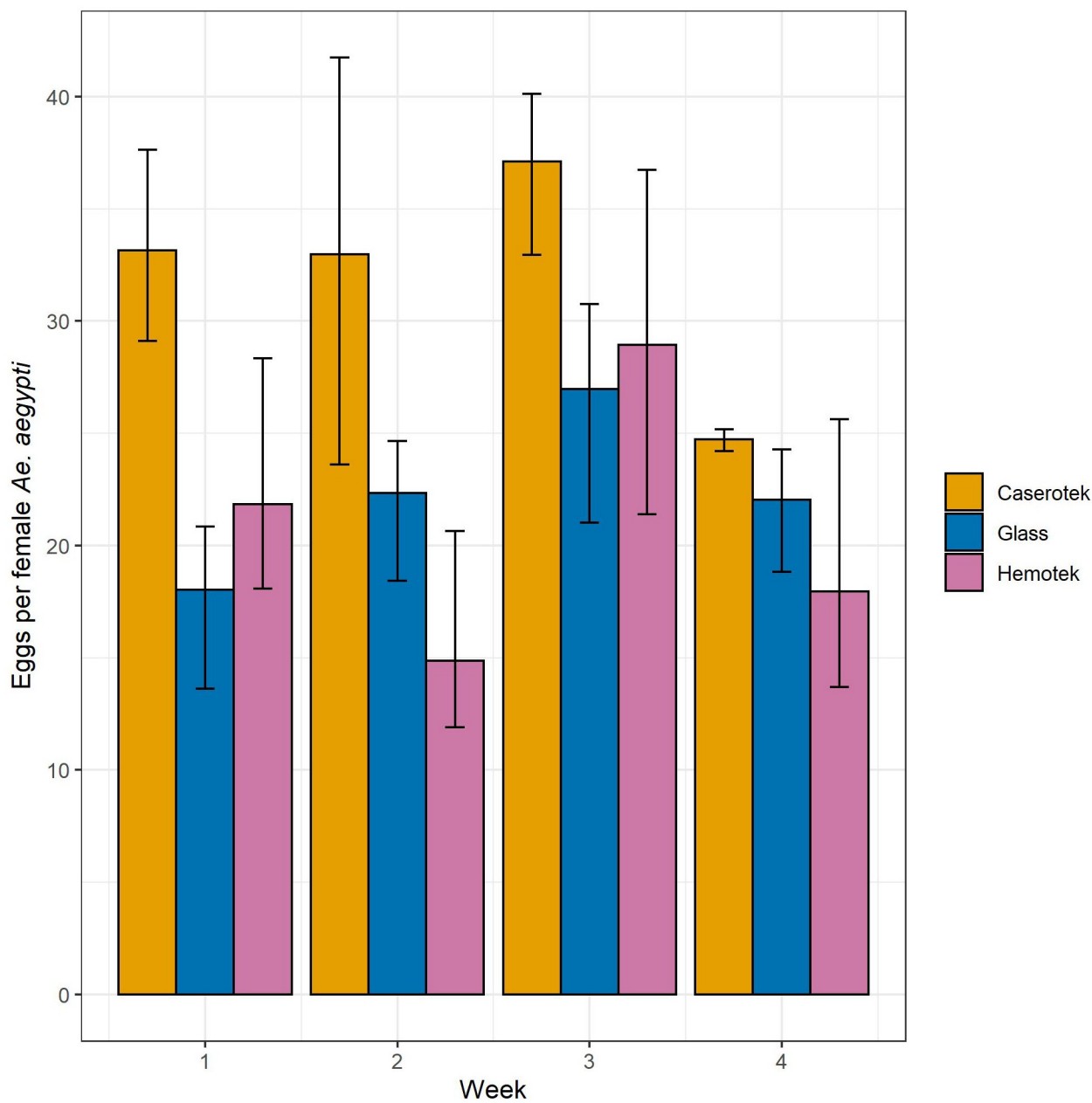

**Fig 5. Eggs per female _Aedes aegypti_ by week and feeding device.** Error bars represent 95% confidence intervals.

Siria device; however, comparison across studies is always difficult because rearing conditions may vary. Both devices were able to keep the blood warm by simply monitoring water temperature and addition more warm water when necessary. The principal difference between the two devices is one is completely disposable whereas the Caserotek device can be reused after washing and sterilization. Additionally, we found the use of PTFE tape to be a major advantage after initially using parafilm as a membrane. Although not widely utilized, PTFE has higher feeding rates than other animal or parafilm membranes in head to head comparisons [13,15].

Interestingly, Caserotek appears to be advantageous when the feeding time was shorter and did particularly well when the mosquitoes were young (first feed in our case). The clear superiority of Caserotek over the other devices during the initial feed suggest that there is some characteristic of this device which stimulates blood feeding more quickly than other devices. It is also notable, that even when additional time for feeding is provided, the Hemotek device had lower feeding rates than the other two devices.

Caserotek was easy to construct with local materials available in Iquitos, Peru and inexpensive (~$5.00 per device). Moreover, the device was easy to use and practical. In resource poor environments, it can be used in circumstances where a slaughterhouse or abattoir is available for blood and can be used where electricity is not available. The only requirement is a thermometer and the ability to boil water. From 2015 to 2019, Caserotek was used to ensure the availability of 500 female *Ae. aegypti* per week for use in direct feeding experiments to study vector competence in active dengue cases [16].

The scalability of Caserotek for mass rearing of mosquitoes or for work with pathogen infected blood would require additional testing. We estimate that a single person could easily manage 30 devices simultaneously, on 30 cages simultaneously by staggering addition of blood and hot water to each device. Hot water would be replaced at 30-minute intervals. We do not recommend Caserotek for producing mosquitoes for large scale release programs, rather the device could facilitate colony maintenance in most environments. Use for vector competence experiments using pathogen infected blood would likely require disposing of the device after use and analysis of the cost efficiency of doing so.

## Study limitations

*Aedes aegypti* experiments were conducted in the UC Davis field laboratory did not have strict temperature and humidity controls. Although the facility had an air conditioner, the facility did not have good thermoregulation. The UC Davis Insectary is representative of facilities on low resource environments and in many ways typical of the indoor environment *Ae. aegypti* thrives in within this large city endemic for dengue transmission [17–20]. Thus, evaluation of these devices under real world conditions is justified. Experiment #1 results were limited by removal of unfed mosquitoes from further inclusion leading to an unbalanced comparison among blood feeding devices during the day 7 and 14 time points in our case but a study design used by others [4]. Another limitation was not evaluating survival in the *Anopheles* experiments.

## Conclusion

Caserotek represents a practical, cost-effective artificial blood feeding device for use in resource poor environments. It has been proven effective for mosquito production and has the potential to use with pathogen infected blood for vector competence experiments.

## Copyright statement

Some authors are military service members and employees of the United States Government. This work was prepared as part of their official duties. Title 17 U.S.C. 105 provides that 'Copyright protection under this title is not available for any work of the United States Government. Title 17 U.S.C. 101 defines a U.S. Government work as work prepared by a military service member or employee of the U.S. Government as part of that person's official duties.

## Disclaimer

The views expressed in this article reflect the results of research conducted by the authors and do not necessarily reflect the official policy or position of the Department of the Navy, Department of Defense, nor the United States Government.

## Supporting information

**S1 Fig. Preparation of the device Caserotek for artificial blood feeding of** *Aedes aegypti* **and** *Anopheles darlingi*.
(TIF)

**S2 Fig. Artificial blood feeding of** *Aedes aegypti* **and** *Anopheles darlingi* **with the Caserotek blood feeding device.**
(TIF)

**S3 Fig. Artificial blood feeding of** *Aedes aegypti* **and** *Anopheles darlingi* **using Hemotek.**
(TIF)

**S4 Fig. Artificial blood feeding of** *Aedes aegypti* **and** *Anopheles darlingi* **with Glass membrane feeders.**
(TIF)

**S5 Fig. Study design for Experiment #1.**
(JPG)

**S6 Fig. Study design for Experiment #2.**
(JPG)

**S7 Fig. Study design for Experiment #3.**
(JPG)

**S8 Fig. Survival curves for** *Aedes aegypti* **after feeding on blood for 30 minutes.**
(TIFF)

**S1 File. Standard operating procedures and maintenance of** *Aedes aegypti* **colonies.**
(PDF)

**S2 File. General Linear Models output for Experiments #1-#3.**
(PDF)

**S3 File. Primary data sets for feeding rate, survival, and egg production analysis in excel format.** Data file includes individual tabs for each type of analysis as well as a data dictionary.
(XLSX)

## Author Contributions

**Conceptualization:** Helvio Astete, Verónica Briesemeister, Cesar Campos, Angel Puertas.

**Data curation:** Helvio Astete, Cesar Campos, Víctor López-Sifuentes, Karin Escobedo-Vargas.

**Formal analysis:** Ryan Larson, Amy C. Morrison.

**Funding acquisition:** Thomas W. Scott, Amy C. Morrison.

**Investigation:** Helvio Astete, Verónica Briesemeister, Cesar Campos, Angel Puertas, Víctor López-Sifuentes, Karin Escobedo-Vargas.

**Methodology:** Helvio Astete, Verónica Briesemeister, Cesar Campos, Angel Puertas, Amy C. Morrison.

**Project administration:** Thomas W. Scott, Michael Fisher, Amy C. Morrison.

**Supervision:** Helvio Astete, Víctor López-Sifuentes, Gissella M. Vásquez, Amy C. Morrison.

**Writing – original draft:** Helvio Astete, Ryan Larson, Gissella M. Vásquez, Amy C. Morrison.

**Writing – review & editing:** Helvio Astete, Verónica Briesemeister, Thomas W. Scott, Víctor López-Sifuentes, Ryan Larson, Michael Fisher, Gissella M. Vásquez, Karin Escobedo-Vargas, Amy C. Morrison.

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
