## [Decision Letter · Decision Letter 0]

6 Jun 2023

Dear Dr. Morrison,

Thank you very much for submitting your manuscript "Evaluation of "Caserotek" a low cost and effective artificial blood-feeding device for mosquitoes." for consideration at PLOS Neglected Tropical Diseases. As with all papers reviewed by the journal, your manuscript was reviewed by members of the editorial board and by several independent reviewers. The reviewers appreciated the attention to an important topic. Based on the reviews, we are likely to accept this manuscript for publication, providing that you modify the manuscript according to the review recommendations. 

Sincerely,

Roberto Barrera, Ph.D.

Academic Editor

Audrey Lenhart

Section Editor

Reviewer's Responses to Questions

**Key Review Criteria Required for Acceptance?**

**Methods**

-Are the objectives of the study clearly articulated with a clear testable hypothesis stated?

-Is the study design appropriate to address the stated objectives?

-Is the population clearly described and appropriate for the hypothesis being tested?

-Is the sample size sufficient to ensure adequate power to address the hypothesis being tested?

-Were correct statistical analysis used to support conclusions?

-Are there concerns about ethical or regulatory requirements being met?

Reviewer #1: (No Response)

Reviewer #2: See general comments below.

**Results**

-Does the analysis presented match the analysis plan?

-Are the results clearly and completely presented?

-Are the figures (Tables, Images) of sufficient quality for clarity?

Reviewer #1: (No Response)

Reviewer #2: See general comments below.

**Conclusions**

-Are the conclusions supported by the data presented?

-Are the limitations of analysis clearly described?

-Do the authors discuss how these data can be helpful to advance our understanding of the topic under study?

-Is public health relevance addressed?

Reviewer #1: (No Response)

Reviewer #2: See general comments below.

**Editorial and Data Presentation Modifications?**

Reviewer #1: (No Response)

Reviewer #2: (No Response)

**Summary and General Comments**

Reviewer #1: In this submission, the authors describe and evaluate a low-tech, affordable blood-feeding device that they developed and have been using in an operational setting for several years already. The device, Caserotek, is constructed of common off-the-shelf products and can be run with minimal infrastructure requirements making the device desirable for low-resource settings. The authors conducted a series of experiments to benchmark the Caserotek against two current gold-standard blood-feeding devices. Overall, the Caserotek performed very well providing evidence that this device could provide a robust alternative to the more resource intensive existing options. The experiments are simple and robust, the analysis is appropriate, and the paper is well written. 

Minor comments: 

Spanish spelling of ‘mosquitoes’ in a number of locations. Need to check and use English spelling for this journal.

The small sample sizes per replicate resulting from removal of unfed females in earlier feedings makes it difficult to interpret later feedings in exp 1. I appreciate that the authors reveal and acknowledge this issue in the results description. Since the analysis is based on rates and percentages, small sample sizes limit resolution of the variability. I suggest removing any data points based on less than 10 females. 

Figure 3: Please make the points larger and consider removing the black border for points and lines. As is, it is very difficult to see the colors for the points, and the color of the lines is obscured by the border. 

The authors make reference to availability of the full model results, but I could not find the model output in the main doc or supplementary files.

Reviewer #2: Authors are suggested to provide information on how much blood (or the minimum amount of blood to be used) to be filled into their device for each feeding, and how long blood can be maintained at 37 °C during the feeding. It would also be helpful to provide a standard operation protocol which can result in over 70% blood feeding rate when using their new device, including the information on mosquito age and feeding time et al. It is unclear how this device can be scaled up to large production facilities, which needs to be discussed. 

Line 287, Survival curves should be provided, with an appropriate statistical analysis, to support their conclusion. 

Line 196, The manufacture and dimension of glass membrane feeders should be provided.

Line 212, when oviposition sites are provided to mosquitoes?

PLOS authors have the option to publish the peer review history of their article (what does this mean?). If published, this will include your full peer review and any attached files.

Reviewer #1: No

Reviewer #2: No

Figure Files:

Data Requirements:

Reproducibility:

References

---

## [Editor Report · Decision Letter 1]

31 Jul 2023

Dear Dr. Morrison,

We are pleased to inform you that your manuscript 'Evaluation of "Caserotek" a low cost and effective artificial blood-feeding device for mosquitoes.' has been provisionally accepted for publication in PLOS Neglected Tropical Diseases.

Best regards,

Roberto Barrera, Ph.D.

Academic Editor

Audrey Lenhart

Section Editor

<style type="text/css">p.p1 {margin: 0.0px 0.0px 0.0px 0.0px; line-height: 16.0px; font: 14.0px Arial; color: #323333; -webkit-text-stroke: #323333}span.s1 {font-kerning: none

</style>

---

## [Editor Report · Acceptance letter]

21 Aug 2023

Dear Dr. Morrison,

We are delighted to inform you that your manuscript, "Evaluation of "Caserotek" a low cost and effective artificial blood-feeding device for mosquitoes.," has been formally accepted for publication in PLOS Neglected Tropical Diseases.

Best regards,

Shaden Kamhawi

co-Editor-in-Chief

Paul Brindley

co-Editor-in-Chief
